# Clinical Presentation and Severity of SARS-CoV-2 Infection Compared to Respiratory Syncytial Virus and Other Viral Respiratory Infections in Children Less than Two Years of Age

**DOI:** 10.3390/v15030717

**Published:** 2023-03-09

**Authors:** Francesco Nunziata, Simona Salomone, Andrea Catzola, Marco Poeta, Federica Pagano, Liana Punzi, Andrea Lo Vecchio, Alfredo Guarino, Eugenia Bruzzese

**Affiliations:** 1Department of Translational Medical Science, Section of Pediatrics, University of Naples Federico II, 80131 Naples, Italy; franc.nunziata@gmail.com (F.N.); simona.salomone94@gmail.com (S.S.); po3ta.89@gmail.com (M.P.); paganofederica1405@gmail.com (F.P.); punziliana@gmail.com (L.P.); andrea.lovecchio@unina.it (A.L.V.); alfguari@unina.it (A.G.); 2Pediatric Unit, OORR Area Stabiese, Castellammare di Stabia, 80053 Naples, Italy; andreacatzola@gmail.com

**Keywords:** COVID-19, bronchiolitis, children, SARS-CoV-2, respiratory syncytial virus

## Abstract

The spread of severe acute respiratory syndrome coronavirus 2 (SARS-CoV-2) and the implementation of restrictive measures led to a dramatic reduction in respiratory syncytial virus (RSV) occurrence together with rare and mild bronchiolitis induced by SARS-CoV-2. We described the respiratory picture of SARS-CoV-2 infection and evaluated the frequency and the severity of SARS-CoV-2 bronchiolitis comparing it with other respiratory viral infections in children less than two years of age. The severity of respiratory involvement was evaluated based on the need for oxygen therapy, intravenous hydration, and the length of hospital stay. A total of 138 children hospitalized for respiratory symptoms were enrolled: 60 with SARS-CoV-2 and 78 with RSV. In the group of SARS-CoV-2-infected children, 13/60 (21%) received a diagnosis of co-infection. Among the enrolled children, 87/138 (63%) received a diagnosis of bronchiolitis. The comparative evaluation showed a higher risk of the need for oxygen therapy and intravenous hydration in children with RSV infection and co-infection compared to children with SARS-CoV-2 infection. In the children with a diagnosis of bronchiolitis, no differences in the main outcomes among the groups were observed. Although children with SARS-CoV-2 infection have less severe respiratory effects than adults, the pediatrician should pay attention to bronchiolitis due to SARS-CoV-2, which could have a severe clinical course in younger children.

## 1. Introduction

The coronavirus disease in 2019 (COVID-19), caused by severe acute respiratory syndrome coronavirus 2 (SARS-CoV-2), is a matter of big concern in global public health. Italy was the first European country affected by the pandemic; from the first cases in March 2020, several waves were observed, resulting in 23 million total cases being registered throughout the four “pandemic waves” [1].

COVID-19 generally has a milder course in children than in adults, with broad clinical manifestations, ranging from asymptomatic infections to mild or moderate illness (fever, headache, cough, vomit, diarrhea, and dyspnea). Severe or critical diseases have rarely been seen, especially in children with underlying medical conditions [2,3].

The percentage of pediatric cases in Italy is 19.7% of the total. Since the beginning of the pandemic, 4.819.122 cases in the population of individuals 0–19 years of age have been diagnosed and reported by the COVID-19 surveillance system of the Italian National Institute of Health (Istituto Superiore di Sanità, ISS), of which 25.389 were hospitalized, 573 were hospitalized in intensive care, and 93 children died [1].

Respiratory syncytial virus (RSV) is a common respiratory pathogen and a leading cause of bronchiolitis among young children [4]. Bronchiolitis is inflammation of the bronchioles usually caused by an acute viral illness. It is the most common lower respiratory tract infection in children younger than 2 years of age. Respiratory distress impedes appropriate oral intake resulting in frequent clinician visits [5]. Bronchiolitis is the most common cause of hospitalization in infants in high-income countries, and RSV is the most common cause of the disease. It is estimated that nearly all children contract the first and most severe RSV infection before reaching 2 years of age and then subsequently experience milder infections later in life [6]. Most children have been infected with RSV by the time they are 2 years of age, and, although most have mild respiratory symptoms, RSV infection can cause severe disease mainly in young children. RSV infection has a substantial global impact, representing the second most frequent cause of death in infants. The transmission of RSV is seasonal, with annual epidemics occurring during the winter months in the northern hemisphere. Like SARS-CoV-2, RSV is primarily transmitted through respiratory droplets (i.e., coughs and sneezes), including indirect contact through contaminated surfaces. Widely implemented non-pharmaceutical interventions (NPIs) against SARS-CoV-2—e.g., stay-at-home orders, the wearing of face coverings, physical distancing, and the promotion of improved hygiene, such as hand washing—all have the potential to prevent the transmission of other communicable (particularly respiratory) diseases. During the SARS-CoV-2 pandemic, a marked decrease in the number of bronchiolitis cases and the disappearance of the RSV winter epidemic were observed, and SARS-CoV-2-related bronchiolitis, although rare, with a mild clinical course, was observed [7]. A significant effect on respiratory and non-respiratory admissions, particularly with decreases in hospital admissions of respiratory infections, was observed with a steep reduction in all RSV indicators, fewer laboratory-confirmed cases, fewer hospital admissions, and fewer instances of emergency department access by children younger than 5 years of age during the pandemic [8,9]. The extraordinary absence of RSV during winter 2020–21 probably resulted in a cohort of younger children without natural immunity to RSV, thereby raising the potential for increased RSV incidence and virulence. During the strict application of NPIs, the overall frequency of community-acquired infections was reduced [10], and after withdrawal of these measures, a rebound rise in some of them (enteroviral infections, bronchiolitis, gastroenteritis, and otitis) was observed whose peaks were beyond the pre-pandemic level [11,12]. In Italy, in the 2021–2022 autumn-winter season, an increase in pediatric hospitalization rates due to respiratory symptoms was observed [13].

In our COVID-19 regional HUB pediatric ward, we observed an increase in moderate respiratory illnesses, which could be caused either by SARS-CoV-2 or by other viral pathogens responsible for similar clinical pictures.

Previous studies reported that the prevalence of viral co-infections associated with SARS-CoV-2 was approximately 10%, with generally poorer clinical outcomes in comparison with isolated SARS-CoV-2 infection. However, viral epidemiology could considerably vary locally, and SARS-CoV-2 may have variable clinical outcomes based on the different variants assessed [14,15]. Li et al., describes the fact that co-infection was relatively common in children with COVID-19. The most frequent co-infected pathogen was mycoplasma pneumoniae (25%) followed by virus (7%) and bacteria 5% co-infection [16]. In a cohort of 93 children with SARS-CoV-2 infection, in 7 (7.5%) patients, co-infection was detected. According to some authors, co-infection is associated with an increase in hospital stays and the worsening of symptoms and complications in patients of older ages (over 65 years). The same study showed that co-infection is more frequent in patients aged 0–5 years (59%) [17]. In a recent systematic review and meta-analysis of pediatric patients, bacterial, fungal, and respiratory viral coinfection rates were 4.73%, 0.98% and 5.41%, respectively. There was increased male predominance, and most of the cases belonged to white (Caucasian) ethnicities. The most common identified virus and bacterium in children with COVID-19 were RSV (31.4%) and mycoplasma pneumonia (23.1%) [18].

Our aims were to describe the clinical respiratory picture of SARS-CoV-2 infection, to compare it with other respiratory viral infections, and to evaluate the frequency and the severity of SARS-CoV-2 bronchiolitis, in children less than two years of age.

## 2. Materials and Methods

### 2.1. Study Design and Population

A retrospective cohort study was performed, collecting data on all children under the age of 2 years, hospitalized between November 2021 and April 2022 for respiratory symptoms in two different hospitals of the Campania region, Italy. Children with SARS-CoV-2 infection and respiratory symptoms were admitted to the COVID-19 regional HUB of the Department of Pediatrics of AOU Federico II in Naples; children hospitalized for RSV infection and negative for SARS CoV-2 infection were enrolled in the pediatric unit of San Leonardo hospital in Castellammare di Stabia (Naples). We included children up to two years of age because the restrictions linked to the COVID19 pandemic during 2020–2021 led to a reduction in the circulation of respiratory viruses in the community, making even older children susceptible to RSV infection. The infections in entire our cohort originate from the Omicron rather than the Delta variant wave. Respiratory symptoms included both symptoms of upper respiratory tract infection (URTI), such as rhinorrhoea and coughs, and signs of respiratory distress (e.g., a high respiratory rate for the patient’s age, the use of an accessory respiratory muscle, intercostal retractions, nasal flaring, crackles or wheezing, and low oxygen saturation levels). The diagnosis of SARS-CoV-2 infection was performed by a specific RT-PCR on a nasopharyngeal swab in all hospital-admitted patients independently of the reason of hospitalization; the diagnosis of other respiratory viral infections was performed, only in presence of respiratory symptoms, by a multiplex PCR on a nasal swab for the following viruses: coronavirus HCoV-NL63, HCoV-OC43, HCoV-229E, HKU1, RSV A-B, rhinovirus, metapneumovirus, influenza A-B, adenovirus, bocavirus, parechovirus, and enterovirus. For each patient, personal and clinical data were collected from medical records. Pre-existing risk factors (such as prematurity, atopy, the type of feeding, having parent who smokes, and comorbidity) for a higher risk of severe illness were evaluated [19].

The children with SARS-CoV2 infection were divided in two groups based on whether or not there was a presence of another viral infection. All the children with co-infection belonged to the group of children who were then referred to a referral center in Naples.

Children with a clinical picture of bronchiolitis, defined according to the literature [20,21,22], were then selected from each group among the patients under the age of two, and the difference in the severity of clinical presentation was evaluated between groups. All children with documented a bacterial etiology or with clinical and laboratory signs of suspected systemic infection were excluded. In detail, we excluded all children with bacterial infection (such as urinary tract infections, sepsis, or gastrointestinal infection in which a bacterial culture was positive); we also excluded all children with documented bacterial pneumoniae (mycoplasma chlamydia, streptococcus pneumoniae or other bacteria detected by a Rt-PCR on a pharyngeal swab). Finally, we excluded children with severe clinical conditions suggestive of sepsis but without a confirmed microbiological diagnosis and children showing a significant increase in inflammatory markers such as severe leukocytosis with significant neutrophilia and increased C-reactive protein (CRP) above 10 times the normal value.

The primary outcomes of our analysis were the evaluation of the length of hospitalization, the need for oxygen therapy, and the need for intravenous hydration; the secondary outcomes were the need for corticosteroid, antibiotic therapy, and inflammatory indexes among the three final groups of children: children with SARS-CoV-2 infection, children with RSV infection, and children with viral co-infection (SARS-CoV-2 plus any other viral respiratory infection). Oxygen supplementation was started when the peripheral oxygen saturation, measured with a pulse-oximeter, was <92%. The oxygen saturation levels used as a guide for commencing supplemental oxygen therapy varied from <90% to <95% among the guidelines. However, the most highly recommended cutoff was <92% [23]. When indicated, oxygen supplementation was performed using a high-flow nasal cannula.

Intravenous fluid administration was started if the child could not ingest enough oral fluids. An inhaled bronchodilator was administered to children with a significant auscultatory finding of wheezing after an initial dose, demonstrating a good clinical response.

We used systemic steroids only if the child’s symptoms showed no improvement or worsened 2 days after starting oxygen supplementation. Finally, we started antibiotic treatment only if the child, during hospitalization, showed an increase in CRP or an X-ray suggestive of bacterial over-infection.

The study was conducted in accordance with the Declaration of Helsinki and approved by the Ethics Committee for Biomedical Activities, University of Naples Federico II, Naples, Italy (protocol code 226/21). Written informed consent to the use of clinical data was obtained from the parents of all the children involved in the study.

### 2.2. Statistical Analysis

The original dataset was created and managed using Microsoft Excel^®^. Differences between groups were evaluated by the Chi-square test or the Fisher’s exact test, when appropriate, for categorical variables and by the non-parametric Mann–Whitney or one-way analysis of variance (ANOVA) for continuous variables, when appropriate. The relative risk for the main severity outcomes was also calculated. Statistical analyses were performed with IBM SPSS Statistics for Windows, Version 26.0 (Armonk, NY, USA: IBM Corp). The statistical significance level was set at *p* < 0.05.

## 3. Results

### 3.1. Study Population and Prevalence of Bronchiolitis

A total of 138 children below two years of age, hospitalized for respiratory symptoms between November 2021 and April 2022, were enrolled. Seventy-eight/138 (56.5%) had RSV infection and 60/138 (43.5%) children had a diagnosis of SARS-CoV-2. In the group of SARS-CoV-2-infected children, 13/60 (21%) received a diagnosis of co-infection showing at least one respiratory virus other than SARS-CoV-2 through a nasal swab. Among the 138 children, 87(63%) received a diagnosis of bronchiolitis (Figure 1).

In the group of co-infected children, the most frequently observed virus was RSV, which was in 8 out of 13 children. Metapneumovirus and bocavirus were observed in two children, rhinovirus/enterovirus in two other children, and coronavirus OC 43 in one child. The main features of the enrolled children are shown in Table 1.

Significant differences were observed in terms of age, between the RSV and co-infected children (mean age 4.5 ± 4.2 versus 7.3 ± 5.6 months), in terms of enteral feeding between the COVID-19- and RSV-infected children (23.4% versus 57.7 % formula feeding), and in terms of prematurity history between the RSV and co-infected subjects (6.4% versus 23.1%). No significant differences between the groups were observed in terms of risk factors for respiratory infections.

### 3.2. Clinical Features

Fevers were significantly more frequently observed in children with SARS-CoV2 infection compared to children with RSV infection (60% versus 9%, respectively; *p* < 0.001), whereas respiratory distress was significantly higher in children with RSV infection compared to children with SARS-CoV2 infection alone (87.2% versus 21.3%, respectively; *p* < 0.001) and in the group of children with co-infection compared to the group with SARS-CoV2 infection alone (69.2% versus 21.3%, respectively; *p* 0.001). No difference in terms of distress was observed between the RSV infected children and co-infected children (87.2% versus 69.7%; *p* 0.085). The distribution of all other symptoms (cough, laryngeal stridor, and gastrointestinal symptoms) was similar between the groups (Table 1).

### 3.3. Biochemical and Radiological Findings

No significant differences were observed in the laboratory findings between the groups except for an average neutrophil count which was significantly lower in COVID-19 children than in RSV-infected or co-infected children (3.30 ± 2.54/mm^3^ versus 5.28 ± 3.36 versus 5.63 ± 3.07, respectively; *p* = 0.002). No significant differences were observed between the groups in inflammatory markers, even if in co-infected children the percent of children with an increase in CRP (>5 mg/L) was higher than in COVID-19- and RSV-infected children (38% versus 36.2% versus 19.2%; *p* = 0.071).

Chest radiography was performed on all children and no difference in the frequency of radiographic abnormalities (interstitial disease, thickening, and lobar findings) between the groups were found.

### 3.4. Difference in the Treatment among Children with SARS-CoV-2, RSV, and Co-Infection

A total of 92% of children with RSV infection, 61.5% of children with co-infection, and only 12.8% of children with SARS-CoV2 infection received systemic steroids. Inhaled bronchodilators were administered to 85.9% of patients with RSV infection, 46.2% of children with co-infection, and only 10.6% of children with SARS-CoV-2 infection.

Overall, therefore, the number of children who required systemic steroids was significantly higher in the group of children with RSV infection. (Table 1).

A total of 54/138 (39%) children received antibiotics, of whom 37 showed increased C-reactive protein amounts suggestive of bacterial over-infection, and 28 of whose X-rays showed thickening suggestive of bacterial infection. Twenty-five children showed both increased C-reactive protein amounts and X-ray thickening suggestive of bacterial pneumoniae.

The need for an antibiotic was significantly higher in children with RSV infection compared to SARS-CoV-2 children. Additionally, children with coinfection were more frequently treated with an antibiotic compared to those with SARS-CoV-2 infection, whereas no significant differences were observed between the group of co-infected children and the group of RSV-infected children, regardless of the hospital to which they were admitted (Table 1).

The need for bronchodilators was higher in children with RSV infection and co-infection compared to children with isolated SARS-C-V-2 infection (Table 1).

### 3.5. Severity Outcomes of Respiratory Involvement among Children with SARS-CoV-2, RSV, and Co-Infection

The management approach was linked to the severity of clinical presentation. We considered oxygen supplementation, intravenous fluid requirement, and the length of hospital stay the severity outcomes. All the outcomes were worse in co-infected and RSV children compared to children with SARS-CoV-2 (Figure 2 a,b). A total of 44/138 (32%) children needed oxygen supplementation with a mean duration of 4.18 ± 1.5 days. A total of 58/138 (42%) children needed intravenous hydration with a mean duration of 3.2 ± 1.5 days. In detail, oxygen supplementation was needed by 53.8% of children with co-infection, 43.6% of children with RSV infection, and only in 6.4% of children with isolated SARS-CoV-2 infection. A significantly higher number of children with RSV and co-infection required parenteral fluid supplementation compared to children with SARS-CoV2 infection. (Figure 2 a,b). No difference in the need for oxygen supplementation was observed between co-infected children and children with RSV infection (RR 1.24 (0.7 to 2.1) *p* = 0.7) (Figure 2c).

Similar results were obtained for intravenous fluid administration. A total of 58/138 (42%) children needed intravenous hydration with a mean duration of 3.2 ± 1.5 days. A total of 55% of children with co-infection, 76.9% of children with RSV infection, and only 12.8% of children with SARS-CoV-2 infection needed the parenteral support of fluids. A significantly higher number of children with RSV and co-infection required parenteral fluid supplementation compared to children with SARS-CoV2 infection. (Figure 2 a,b). No difference in the need for parenteral fluids was observed between co-infected children and RSV infected children [RR 1.26 (0.8 to 1.9) *p* = 0.5] (Figure 2c).

According to an ANOVA test, the length of hospital stay among the three groups was not statistically significant; however, the mean duration of hospital stay was significantly higher for co-infected children compared to children with SARS-CoV-2 infection, (6.0 ± 2.4 vs. 4.6 ± 1.9 days; *p* 0.01) (Figure 2b). No difference was observed between co-infected children and RSV-infected children (6.0 ± 2.4 vs. 5 ± 2.5 days; *p* = 0.152) (Figure 2c).

### 3.6. Clinical Characteristics and Severity of Bronchiolitis in Children with SARS-CoV-2, RSV, and Co-Infection

Finally, we evaluated the outcomes in children with a clinical diagnosis of bronchiolitis in the three groups. The number of children with bronchiolitis was significantly lower in the group of children with SARS-CoV-2 infection compared to the children with bronchiolitis due to RSV infection or viral co-infections (10/47 (21.3%) versus 68/78 (87.1%) and versus 9/13 (69.2%), respectively; *p* < 0.001).

Fever was more frequently observed in children with SARS-CoV-2 bronchiolitis compared to children with RSV bronchiolitis and compared to children with bronchiolitis and viral co-infections (60% versus 13.2% versus 44.4%, respectively; *p* = 0.001).

Coughing and poor feeding were significantly more common in children with bronchiolitis due to RSV infection compared to children with SARS-CoV2 bronchiolitis (60.3% versus 20%; *p* < 0.001 and 67.2 % versus 20%; *p* < 0.05, respectively) suggesting an increased severity of respiratory involvement in RSV infection. No significant difference in the frequency of other symptoms was observed among the groups.

For the laboratory evaluation, no major differences were observed even if in RSV infected children the mean number of white blood cells (WBC) was higher than that in COVID-19-infected children (13.4 ± 5.4/mm^3^ versus 9.5 ± 1.5/mm^3^; *p* = 0.028).

To evaluate whether SARS-CoV-2 bronchiolitis was more or less severe than bronchiolitis induced by other viral infections, we analyzed the main clinical outcomes of severity among the three groups of children. No statistical differences in the relative risk were observed among the three groups in the length of hospital stay and oxygen requirement, whereas the requirement of parenteral infusion was significantly increased in the children with RSV infection compared to those with SARS-CoV-2 infection. (Figure 3a–c). Again, bronchodilators and systemic steroids were needed in a significatively higher percentage of children with RSV bronchiolitis compared to children with SARS-CoV-2 bronchiolitis (83.8% vs. 50%; *p* = 0.026 and 91.2% vs. 30%; *p* < 0.001, respectively).

Only two children had symptoms that worsened during hospitalization and were admitted to PICU, one child developed mild pneumothorax, and one child died. All children were in the group of RSV infection and were younger than 6 months old.

## 4. Discussion

Bronchiolitis is a potentially severe respiratory presentation of acute respiratory infections, induced by viruses, namely RSV, influenza, and others in infants and younger children. Although COVID-19 in children is generally a mild respiratory infection, it may present respiratory distress like another viral bronchiolitis [24]. A recent report suggests that RSV-infected patients require a higher level of medical care and have to stay in hospital for longer than SARS-CoV-2-infected children [25].

In our study, we included patients admitted for respiratory symptoms with SARS-CoV-2 infection and compared them with patients with respiratory symptoms, RSV infection, and co-infection (SARS-CoV-2 and any other respiratory viruses). During the period between November 2021 and April 2022, 87 children under two years of age were admitted to our center, for SARS-CoV-2 infection. A total of 60/87 (69%) presented respiratory symptoms. This percent is quite high, probably due to the high prevalence of the Omicron variant, which was the main circulating variant of SARS-CoV-2 in Italy in that period. The comparative clinical evaluation confirms, as reported in the literature [25], that patients with SARS-CoV-2 infection have milder respiratory symptoms and a shorter duration of hospitalization compared to patients with RSV infection or co-infection. This difference was observed in all the main outcomes of severity, supporting the hypothesis that the severity of symptoms in the group of children with co-infection was related to the presence of RSV rather than SARS-CoV-2.

Furthermore, the need for adjunctive drugs, such as systemic steroids, bronchodilators, and antibiotics, was less in the SARS-CoV-2 patients compared to the other two groups of children. These differences in the use of steroids and antibiotics may be perhaps interpreted as being due to the different management practices in different hospital settings. These differences could, at least in part, be attributed to different management practices in different hospitals. However, both antibiotics and systemic steroids were prescribed more frequently to the co-infected and RSV-infected children, compared to SARS-CoV-2 children, regardless of the hospital to which they were admitted. Co-infected children were managed at the COVID-19 regional HUB of the Department of Pediatrics of AOU Federico II in Naples, whereas RSV infected children were exclusively managed at the pediatric unit of San Leonardo hospital in Castellammare di Stabia (Naples). This suggests that different management practices were not able to influence the results. Overall, these data showed that patients infected with SARS-CoV-2 show mild respiratory effects compared to children with RSV infection and co-infection (SARS-CoV-2 and another respiratory virus).

To assess the differences in the clinical severity of bronchiolitis due to SARS-CoV-2, we selected all patients with a clinical diagnosis of bronchiolitis and analyzed the main clinical outcomes of severity among children with a different etiology. About 60% of the entire population of enrolled children fulfilled the criteria for the clinical diagnosis of bronchiolitis (as defined in the Methods section). In our population, bronchiolitis was more frequently observed in the group of children with RSV infection 68/78 (87.1%) than in children with co-infection 9/13 (69.2%) or children with SARS-CoV-2 infection alone 10/47 (21.3%). This result agrees with data from the literature showing that bronchiolitis due to SARS-CoV-2 is less frequently observed than RSV-induced bronchiolitis is [7]. Despite this, the incidence in our population is still higher than that reported in the literature, also because the infections of our cohort originate from the Omicron rather than the Delta variant wave. The Omicron variant is more infectious, and it is described to cause a higher symptom burden in children compared with children with other variants and with adults, possibly due to previous vaccination [26]. The Omicron variant was associated with an increase in respiratory symptoms compared with the wild type/Alpha variant, with coughing, fever, sore throat, nasal congestion/runny nose, and fatigue being the more frequently described symptoms in children [26]. Finally, recent studies indicate disproportionately higher hospitalization rates in children after the emergence of Omicron [27,28] with more severe complications affecting the neurological and respiratory systems [29].

Surprisingly, in terms of severity, children with SARS-Cov2 bronchiolitis have a similar, not better, clinical course of disease compared to children with RSV and co-infection bronchiolitis. Notably, they did show statistically significant differences in the length of hospital stay and oxygen requirements. A significant difference was observed in the risk of systemic steroids, inhalers bronchodilators, and parenteral hydration being required, with RSV bronchiolitis having a higher risk for all the above-mentioned interventions. Again, this additional risk may likely be due to the different practices in the management of children with acute respiratory infection in different hospitals. However, no significant differences were observed between the group of co-infected children and the group of RSV-infected children in the requirement for steroids, bronchodilators, and antibiotics, regardless of the hospital, suggesting that the different management practices were not able to influence the results.

The study has some limitations, of which the involvement of two centers in the same city is one, which did not allow the generalization of the results, the possible effects of the different practices of management of acute respiratory diseases in different hospitals is another, and the small sample size that neither allowed us to confirm the lower severity of bronchiolitis from SARS-CoV-2 nor allowed us to estimate its real incidence is one. Despite the limitations, these data showed that, in children under two years of age with SARS-CoV-2 infection, respiratory symptoms are less severe compared to those observed in children with RSV infection or co-infection; in addition, SARS-CoV-2 infection is confirmed to be a less common cause of bronchiolitis compared to SARS-CoV-2. Surprisingly, despite what reported in the literature [7] our data showed that SARS-CoV-2 bronchiolitis is not mild. Our cohort of children with SARS-CoV-2-induced bronchiolitis showed a similar risk of a severe clinical course compared to children with bronchiolitis induced by RSV and co-infections. Similarly, in our population, the presence of viral co-infection was not associated with an increased risk of worse outcomes [17,18], in-keeping with the findings by Halabi et al. [30] that account for a possible “negative” viral interference between SARS-CoV-2 and RSV. Furthermore, this discrepancy may also be explained by the small number of children with co-infection, and needs to be resolved by studies with a higher sample size.

## 5. Conclusions

In conclusion, although SARS-CoV-2 infection in people of a pediatric age is generally milder than in adults, respiratory effects in children also appear frequently when infected with the Omicron variant wave. SARS-CoV-2 should be included in the list of viral pathogens responsible for bronchiolitis. Clinical attention should be paid to patients presenting more severe respiratory symptoms for determining the risk of co-infection with other respiratory viruses. Finally, children with SARS-CoV-2 bronchiolitis may have a severe clinical course like that of children with RSV bronchiolitis.

## Figures and Tables

**Figure 1 viruses-15-00717-f001:**
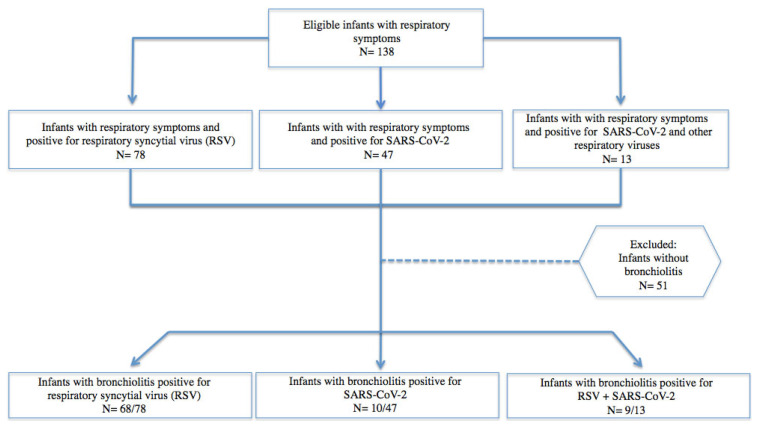
Eligible patients under two years of age with respiratory symptoms and bronchiolitis due to SARS-CoV-2, RSV, and viral co-infection.

**Figure 2 viruses-15-00717-f002:**
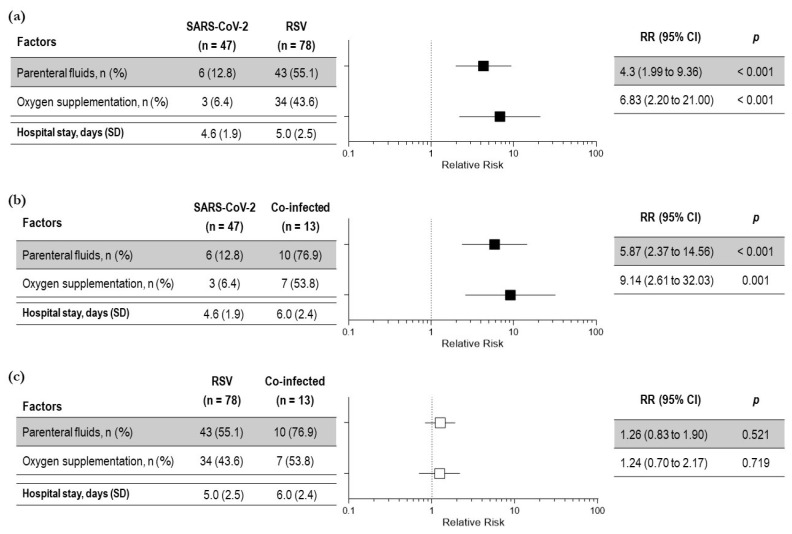
Relative risk of severity outcomes among children with respiratory symptoms due to SARS-CoV-2 and RSV (**a**); SARS-CoV-2 infection and co-infection (**b**); RSV infection and viral co-infection (**c**)—total number of children: 138. Black squares indicate significant results; white squares indicate insignificant results.

**Figure 3 viruses-15-00717-f003:**
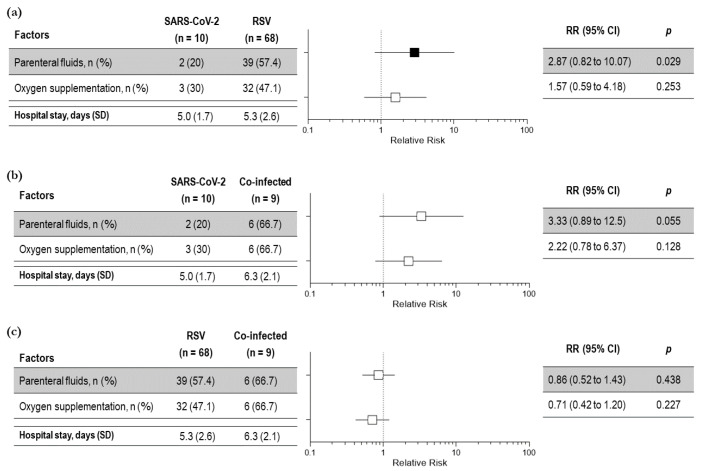
Severity outcomes among children with bronchiolitis due to SARS-CoV-2 and RSV (**a**); SARS-CoV-2 infection and co-infection (**b**) and RSV infection and viral co-infection (SARS-CoV-2 and RSV) (**c**)—total number of children: 87. Black squares indicate significant results; white squares indicate insignificant results.

**Table 1 viruses-15-00717-t001:** Main features of enrolled children.

	Overall (*n* = 138)	SARS-CoV-2 (*n* = 47; 34%)	RSV (*n* = 78; 56.6%)	Co-Infection (*n* = 13; 9.4%)	*p*
**Clinical features**					
Fever, *n* (%)	55 (39.9)	37 (78.7)	6 (46.2)	12 (15.4)	<0.001
Respiratory distress, *n* (%)	87 (63)	10 (21.3)	9 (69.2)	68 (87.2)	<0.001
Cough, *n* (%)	74 (53.6)	13 (27.7)	6 (46.2)	55 (70.5)	<0.001
Rhinorrea, *n* (%)	79 (57.2)	10 (21.3)	6 (46.2)	63 (80.8)	<0.001
Laryngeal stridor, *n* (%)	10 (7.2)	2 (4.3)	2 (15.4)	6 (7.7)	0.381
Poor feeding, *n* (%)	68 (49.3)	16 (34)	5 (38.5)	47 (60.3)	0.013
GI symptoms, *n* (%)	14 (10.1)	8 (17)	0 (0)	6 (7.7)	0.110
Neurological symptoms, *n* (%)	6 (4.3)	4 (8.5)	1 (7.7)	1 (1.3)	0.131
Laboratory findings					
WBC (10^3^ cells/μL), mean (SD)	11.32 (5.01)	8.77 (3.37)	12.8 (5.26)	12.06 (5.1)	<0.001
Neutrophils (10^3^ cells/μL), mean (SD)	4.61 (3.19)	3.30 (2.54)	5.28 (3.36)	5.63 (3.07)	0.002
Lymphocytes (10^3^ cells/μL), mean (SD)	5.35 (2.79)	4.4 (2.0)	6.04 (3.12)	5.02 (2.41)	0.006
CRP, mg/L mean (SD)	14.3 (28.2)	6.8 (10.6)	22.8 (17.4)	32.1 (61.6)	0.007
Therapies					
Antibiotics, *n* (%)	54 (39.1)	10 (21.3)	38 (48.8)	6 (50)	0.008
Parenteral hydration, *n* (%)	56 (42.7)	6 (12.8)	43 (55.1)	10 (76.9)	<0.001
Systemic Steroids, *n* (%)	86 (62.3)	6 (12.8)	72 (92.3)	8 (61.5)	<0.001
Bronchodilators, *n* (%)	78 (56.5)	5 (10.6)	67 (85.9)	6 (46.2)	<0.001

## Data Availability

The datasets generated and/or analyzed during the current study are available from the corresponding author on reasonable request.

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
