# Peer review of "Clinical Presentation and Severity of SARS-CoV-2 Infection Compared to Respiratory Syncytial Virus and Other Viral Respiratory Infections in Children Less than Two Years of Age"

_viruses, 2023, doi:10.3390/v15030717_

Round 1

Reviewer 1 Report

Review Nunziata et al.

Nunziata and colleagues performed a retrospective cohort study collecting data of all children under the age of 2 years, that had been hospitalized between November 2021 and April 2022 for respiratory symptoms in two different hospital settings.

The aim of the here described study was to describe the respiratory picture of SARS-CoV-2 infection and to evaluate frequency and severity of SARS-CoV-2 bronchiolitis in comparison to other respiratory viral infections or co-infections in this specific age group of children. Among the enrolled children, 87/138 (63%) received a diagnosis of bronchiolitis. The comparative evaluation showed a higher risk of oxygen therapy, intravenous hydration, for systemic corticosteroids and antibiotics in children with RSV infection and co-infection compared to children with SARS-CoV-2 infection. In children with clinical diagnosis of bronchiolitis no difference in terms of main outcomes among groups were observed and as such the authors advise paediatricians to pay attention to SARS-CoV-2 induced bronchiolitis that may have a severe clinical course.

This is a diligently prepared manuscript, which indeed focusses on an important topic. However, I have serious concerns about the general comparability of the 2 groups, see “Majors”, point 3, and ask myself if this manuscript should not solely focus on the children with SARS-CO-2 in comparison to children with SARS-CoV-2 plus a 2nd virus. If the analysis were limited to these 2 groups (which to my understanding were treated in the same hospital) than a comparison would be fair and would not include such a huge “treatment habit bias” (as treatment in 2 different hospitals without doubt is).

Majors:

1.       Methods, Line 104: “Children with clinical picture of bronchiolitis, defined according to literature [12-14]”… When studying the literature, the terms “bronchiolitis” and “children with obstructive bronchitis” are frequently confused or mixed up. Please include a clear definition of both into your draft and verify / clarify whether your patient collective indeed matches this definition. When applying these definitions, do you really have so many children around 2 years of age that have bronchiolitis (i.e. with “crackles or wheeze”)?  

2.       Methods, line 87: “Children with SARS-CoV-2 infection and respiratory symptoms were admitted to the COVID-19 Regional HUB of the Department of Pediatrics of AOU Federico II in Naples”. Does this include the group with co-infection (SARS-CoV-2 plus other virus)? If so, please indicate this. (And if so, these two groups treated in the identical hospital could be reasonably compared).

3.       Fig. 2, line 204: “Relative risk of severity outcomes”. First, this Fig. does not only describe “severity outcomes” (such as LOS, oxygen dependency) but it also describes “modes / regimens of treatment”. Please specify the Figure Legend Heading accordingly.

Second, this Fig. with its subfigures should be divided into “Primary outcome data” and “secondary endpoints” according to the specification given in the Methods section.

Third, my main concern is that the 2 patient groups were treated in different hospitals. As such, the here depicted differences in treatment regimens might well be attributed to different habits / practises but not because of differences caused by the virus itself. (As an example, we would never use inhaled steroids in the acute phase of bronchiolitis whereas in the here presented manuscript 100% of the children indeed received those).

You yourself address this important concern, (p. 8, second paragraph), however what is left of this analysis, when differences (including the most important item “LOS”) are attributed to hospital policies?

4.       Methods, line 102: “All children with documented bacterial etiology or with clinical and laboratory signs of suspected systemic infection were excluded”. Please specify with respect to “documented bacterial infection”: as defined by what? Please specify with respect to “clinical and laboratory signs of suspected infection”: what were those clinical and laboratory markers? Why did almost 50% of the remaining children, that were included into the study receive antibiotics?

5.       Discussion, line 291 ff. “This result agrees with data from literature showing that bronchiolitis due to SARS-CoV-2 is less frequent than RSV induced bronchiolitis [5]”.

Indeed, it is generally assumed that SARS-CoV-2 induced bronchiolitis is comparatively rare. Could you provide the reader with the overall incidence of this entity in your collective? In line with my above-described concern (“Major”, Point 1), does your estimated incidence match published data (or in other words: if you find a comparatively higher incidence, would you question the diagnosis “bronchiolitis” in your cohort?) Is the incidence of SARS-CoV-2 bronchiolitis in your cohort indeed rarer than RSV-induced bronchiolitis? Please discuss.

6.       SARS-CoV-2 Variants: Your cohort originates from the Omicron rather than Delta variant wave. However, neither in the methods nor in the discussion, this point is reflected on. I think that this point should be made clear to the reader (as we are mostly lacking structured data during the Delta variant wave).

Minors:

1.       Table 1, Colum 1: What does “Brotherhood” mean? Is this the correct English term? Would you mean “Sibling”? Please correct.

2.       Under Table 1, line 144: Please correct the spelling mistake (missing s): “Significant differences were observed in terms of age, between RSV and co-infected 144 children (mean age 4.5±4.2 versus 7.3±5.6 months), in term (missing s) of enteral feeding between 145 COVID-19 and RSV infected ….”

3.       Same applies to line 157: “No difference in term of distress was observed between RSV 157 infected children and co-infected children (87.2% versus 69.7%; p 0.085)…”.

4.       Results, line 222: “…whereas poor-feeding was significantly more common in children with bronchiolitis due to RSV infection compared to children with SARS-CoV2 bronchiolitis (60.3% versus 20% respectively; p< 0.001) suggesting an increased severity of respiratory involvement (in which group?)”. This sentence includes so many comparisons that it would be easier if you added this (admittedly redundant) information in end of the sentence.

5.       Discussion: line 284: “Overall, our result showed that patients infected with SARS-CoV-2 have a mild respiratory involvement. (point is missing at the end of the sentence). In order to assess the difference in clinical severity of bronchiolitis due to SARS-Cov-2 or other viruses, we selected all patients with clinical diagnosis of bronchiolitis and analyzed the main clinical outcomes of severity between children with differ (different?, please correct) etiology”.

6.       Discussion, line 288: About 60% of the entire population of enrolled children fulfilled the criteria for clinical diagnosis of bronchiolitis” … I would suggest adding “as defined above in the Methods section”. Please refer to “Majors”, point 1.

7.       Addendum, line 331: “Institutional Review Board Statement: The study was conducted in accordance with the Declaration of Helsinki and approved by the Ethics Committee for Biomedical Activities, University”. Please include information on the consent procedure in the main part of the manuscript, i.e. Methods section. Same applies to “Informed Consent Statement”.

Reviewer 2 Report

In the manuscript entitled ‘’Clinical presentation and severity of SARS-CoV-2 infection compared to Respiratory Syncytial Virus and other viral respiratory infections in children less than two years of age by Fransesco Nunziata et al, the authors present combined data from 2 different hospitals in order to compare severity of infection in 3 different study groups of children under the age of two: mono infection with SARS-CoV-2 or RSV and co-infection of SARS-CoV-2 with other viruses or RSV co-infection as a subgroup. Primary outcomes comprised length of hospital stay, the need of oxygen therapy and intravenous hydration, whereas secondary outcomes were the use of systemic corticosteroids and antibiotics, and the rise of inflammatory markers. Finally, the authors suggest that SARS-CoV-2 bronchiolitis may have a severe clinical course.  

The authors acknowledge the limitations of their study being the small sample size and a bias regarding management. Indeed, management of bronchiolitis varies a lot between centres and even among attending physicians of the same clinic. However, the treatment plan of patients including escalation of treatment as needed, lacks essential details which reflects some flaws in the design of the study. For how many days were the patients treated with supplemental oxygen or iv fluids and were same thresholds applied the at two different centres? What was the clinical threshold to initiate bronchodilators/inhaled or systemic steroids in the different subgroups? In this regard, I would suggest since steroids (inhaled or systemic) is not routinely recommended in the guidelines cited by the authors as a therapeutic strategy it does not seem appropriate to use it as an outcome of the study to describe disease severity. Instead, PICU admission or use of high flow nasal cannula/ positive pressure ventilation which have not been discussed at all in the manuscript, would be more objective and of interest in order to assess clinical severity.

Another debatable secondary outcome is the use of antibiotics as a measure of disease severity. Since patients with bacterial/systemic infection have been excluded, the use of antibiotics may also be a biased practice in view of the lack of significant differences in the values of inflammation markers between the study groups.  

In Figure 1 there is a discrepancy in numbers compared with the text (line 135): As I can see authors use 9 co-infected patients with RSV and SARS-CoV-2 in further statistical analyses (figure 3), as opposed to 8 described in the main text (line 135) which most probably should be corrected.

Finally, authors should better give away the message for the readers instead of the generic message which does not add in medical literature.

Reviewer 3 Report

Abstract

I suggest authors introduce the acronymous at the first writing e.i. SARS-CoV-2 (line 11 page 1), RSV (line 12 page 1) in the abstract

It is unusual for a European-based study to use the USA bronchiolitis age threshold of two years.

As antibiotics and systemic corticosteroids are not treatments for bronchiolitis I strongly disagree with the authors' choice to use these as a severity marker (line 17 page 1).

Is well known that SARS-CoV-2 can present a more severe course in younger patients page 1 lines 24-25 (https://pubmed.ncbi.nlm.nih.gov/32424745/). 

Introduction

I suggest avoiding the term "hardest" hit country because it is difficult to quantify (page 1 line 31). I suggest rephrasing "Severe or critical diseases have been rarely seen, especially in children with underlying medical conditions" because the second part of the sentence sounds the opposite of the intended (page 1 lines 37-38).

I suggest explaining what Istituto Superiore di Sanità means (page 1 line 41).

The term bronchiolitis implies less than 2 or 1 year of age pending definition and the term infant implies under one year of age; I'm a little confused by "young infants" (page 2 line 46). 

The authors refer to pre-pandemic RSV seasonality; I suggest introducing a disclosure about it (page 2 line 58) e.i. https://pubmed.ncbi.nlm.nih.gov/34403164/.

I suggest adding the term virulence after incidence (line 65 page 2).

Page 2 line 70 a reference is needed e.i: https://pubmed.ncbi.nlm.nih.gov/36083314/.

I suggest adding the age range to the aims of the study (page 2 line 79).

The meaning of "two different hospital settings." is unclear (page 2 line 87).

I suggest naming the peripheral hospital of Naples (page 2 line 90).

I suggest mentioning the clinical or management conditions that led to RT-PCR testing for SARS-CoV-2 and multiplex; both are routinely performed on each patient or not? (page 2 lines 95-96).

I suggest adding a citation to page 3 lines 100-101 (https://pubmed.ncbi.nlm.nih.gov/34097050/).

I suggest moving exclusion criteria after the inclusion criteria (page 3 "All children with documented bacterial etiology or with clinical and laboratory signs

of suspected systemic infection were excluded"). 

Is unclear which clinical records the authors reviewed searching for "bronchiolitis clinical picture": clinical records of all patients under the age of two or patients discharged with that diagnosis or something else? (page 3 line 12?)

I suggest adding the version and the logo ® after "Microsoft Excel" (page 3 line 114). 

43.5% hospitalized for COVID-19 seems an extremely high ratio for my experience of that season in Italy;  as mentioned I need more clear inclusion criteria to better evaluate the numbers and I need to know if each patient diagnosed with COVID-19 was hospitalized regardless of symptoms or not (page 3 line 122).

I suggest introducing the ANOVA test in the three-group analysis if appropriate. (116).

I suggest changing the sentence "showing at least one other respiratory virus at nasal swab" to: showing at least one respiratory virus other than SARS-CoV-2 at the nasal swab. (page 3 line 126)

I suggest reporting p value also for fever (page 4 line 152).

To complete the paragraph on biochemical findings I suggest creating a table or adding lines to the clinical feature table with CRP, neutrophils white cells that I consider interesting (page 5 line 165).

page 5 I strongly suggest eliminating the use of antibiotics and steroids from this article as a marker of severity (page 5 lines 175, 176 and tables).

I suggest to add the p value of the length of stay and I suggest to use the log-rank test to asses it (page 5 line 177)

To better understand the article I would like to know the test used here "(60% versus 13.2% versus 44.4% respectively; p=0.001)" (page 6 line 220).

I'm not sure "although" is the correct term in this context (page 6 line 227)

I suggest to specify "in children" after the sentence "is generally a mild respiratory infection" (page 8 line 268)

Check for misspellings "A recent report suggest" -s or -ed (page 8 line 269)

Consider if the term "that" could be more appropriate than "between" (page 8 line 276).

I would avoid the term steroids in this setting (page 8 lines 277-283). There is no evidence of "in the attempt to obtain a 282

faster improvement in general clinical conditions with a consequent faster discharge" or at least this controversial aspect would benefit from an extended literature or a huge discussion that I strongly suggest the authors avoid, similarly for antibiotics (page 8 line 297) and again steroids (page 8 line 298)

"SARS-Cov2" misspelling, the v should be a capital letter (page 8 line 294)

I consider the involvement of two centers in the same city a limitation that did not allow generalizing the results even more than the sample size (page 8 line 304).

The sample size could be addressed for this "our data failed to demonstrate that SARS-CoV-2 bronchiolitis is generally mild." page 8 line 309

Overall I suggest adding citations in the discussion section.

Conclusions: I consider the conclusion a little bit vague and I suggest that there are other and more suitable take-home messages from this study e.i. Search for coinfections in the case of a severe respiratory clinical picture in a SARS-CoV-2 infected subject. 

Round 2

Reviewer 1 Report

Thank for this nice revision of your manuscript. 

You might again want to check for spelling mistakes, wording etc.

However, to me the manuscript has considerably improved.